# Body Mass Index Z-Score Modifies the Association between Added Sugar Intake and Arterial Stiffness in Youth with Type 1 Diabetes: The Search Nutrition Ancillary Study

**DOI:** 10.3390/nu11081752

**Published:** 2019-07-30

**Authors:** Natalie S. The, Sarah C. Couch, Elaine M. Urbina, Jamie L. Crandell, Angela D. Liese, Dana Dabelea, Grace J. Kim, Janet A. Tooze, Jean M. Lawrence, Elizabeth J. Mayer-Davis

**Affiliations:** 1Department of Health Sciences, Furman University, Greenville, SC 29613, USA; 2Department of Rehabilitation, Exercise and Nutrition Sciences, University of Cincinnati Medical Center, Cincinnati, OH 45267, USA; 3The Heart Institute, Cincinnati Children’s Hospital Medical Center and Department of Pediatrics, University of Cincinnati, Cincinnati, OH 45229, USA; 4Department of Biostatistics, Gillings School of Global Public Health, School of Nursing, University of North Carolina, Chapel Hill, NC 27599, USA; 5Department of Epidemiology and Biostatistics, Arnold School of Public Health, University of South Carolina, Columbia, SC 29208, USA; 6Department of Epidemiology, Colorado School of Public Health, University of Colorado Denver, Aurora, CO 80045, USA; 7Department of Pediatrics, University of Washington, Seattle, WA 98195, USA; 8Department of Biostatistics and Data Sciences, Wake Forest School of Medicine, Winston-Salem, NC 27157, USA; 9Department of Research & Evaluation, Kaiser Permanente Southern California, Pasadena, CA 91101, USA; 10Department of Nutrition, Gillings School of Global Public Health, University of North Carolina, Chapel Hill, NC 27599, USA

**Keywords:** added sugar, arterial stiffness, diabetes, epidemiology, adolescents and young adults

## Abstract

The relationship between added sugar and arterial stiffness in youth with type 1 diabetes (T1D) has not been well-described. We used data from the SEARCH for Diabetes in Youth Study (SEARCH), an ongoing observational cohort study, to determine the association between added sugar and arterial stiffness in individuals diagnosed with T1D <20 years of age (*n* = 1539; mean diabetes duration of 7.9 ± 1.9 years). Added sugar intake was assessed by a food frequency questionnaire, and arterial stiffness measures included pulse wave velocity (PWV) and augmentation index. Separate multivariate linear regression models were used to evaluate the association between added sugar and arterial stiffness. Separate interaction terms were included to test for effect modification by body mass index (BMI) z-score and physical activity (PA). Overall, there was no association between added sugar and arterial stiffness (*P* > 0.05); however, the association between added sugar and arterial stiffness differed by BMI z-score (*P* for interaction = 0.003). For participants with lower BMI z-scores, added sugar intake was positively associated with PWV trunk measurements, whereas there was no association for those who had a higher BMI z-score. PA did not significantly modify the association between added sugar and arterial stiffness. Further research is needed to determine the longitudinal relationship and to confirm that obesity differentially affects this association.

## 1. Introduction

Current recommendations from the Dietary Guidelines for Americans and the World Health Organization advise individuals to consume less than 10% of their total caloric intake from added sugars [1], while the American Heart Association recommends that children (2–18 years of age) should consume less than 100 calories of added sugar per day [2]. However, added sugars account for approximately 15% of the total calories consumed in the general population of youth, with the leading source being sugar-sweetened beverages (SSBs) [3,4,5].These findings are concerning given the mounting evidence that added sugar is adversely associated with cardiovascular disease (CVD) [2,6]. Among youth, higher added sugar or SSB intake is associated with lower high-density (HDL) lipoprotein cholesterol [7,8], increased triglyceride levels [8], and higher blood pressure [9], independent of adiposity [10]. However, there are conflicting findings suggesting either a more nuanced relationship exists between added sugar and CVD or that the null findings are due to differences in study designs or the specific CVD risk factor or outcome being examined. For example, added sugar intake was associated with increased CVD mortality in U.S. adults [11] but not in elderly Chinese adults [12]. In youth with type 1 diabetes (T1D), SSB consumption was positively associated with total cholesterol and low-density lipoprotein (LDL) cholesterol [13]. In contrast, among U.S. adolescents, added sugars were associated with HDL cholesterol and triglycerides, but not LDL cholesterol [5].

Arterial stiffness is a measure of subclinical CVD and has been associated with markers of atherosclerosis, stroke and coronary heart disease, and mortality in adult populations [14,15,16,17,18]. However, there have been few studies examining the relationship between added sugar with arterial structure and function. In adults, the effect of SSB or sugar-based products has also yielded mixed findings with some studies showing high SSB adversely affecting endothelial function [19,20,21], while other studies show no effect on arterial stiffness [22]. A clearer understanding of the role of added sugar on CVD and associated risk factors is warranted, especially in youth when the arterial structure and function are still modifiable [23]. Further, determining this relationship in youth with T1D is particularly critical as they are at substantially increased risk of developing CVD at earlier ages [24], and evidence shows that 50% of youth with T1D consume SSBs [13] despite recommendations to limit added sugar intake [25].

We aimed to determine the association between added sugar and arterial stiffness, which is an independent predictor of CVD events and measures subclinical disease [26] in individuals with type 1 diabetes. In addition, given increasing evidence suggesting that obesity and physical activity may modify the effect of nutrition on health outcomes [27,28], we also tested whether the associations between added sugar and arterial stiffness differed by body mass index z-score and physical activity.

## 2. Materials and Methods

### 2.1. SEARCH for Diabetes in Youth Study (SEARCH)

Data for this study were derived from the SEARCH Study and the SEARCH Nutritional Ancillary Study. SEARCH ascertained cases of diabetes among individuals diagnosed before the age of 20 years from 2002 onward. A detailed description of SEARCH study methods has been published elsewhere [29]. Briefly, SEARCH participants who were newly diagnosed in 2002–2006 or 2008 from 5 United States centers (Cincinnati, Ohio and surrounding counties; Colorado with southwestern Native American sites; Kaiser Permanente Southern California members; Seattle, Washington and surrounding counties; and South Carolina) completed a baseline study visit. Participants who were diagnosed in 2002–2005 were also invited for follow-up visits at approximately 12, 24, and 60 months after baseline visit. A subset of SEARCH participants aged 10 years and older who had a least 5 years of diabetes duration were recruited for an additional outcome study visit between 2011 and 2015 to ascertain additional data and measurements of diabetes-related complications and comorbidities [24]. The SEARCH Nutrition Ancillary Study was designed to assess the associations of nutritional factors with the progression of insulin secretion defects and CVD risk factors in youth with T1D. Both studies were reviewed and approved annually by the local institutional review boards that had jurisdiction over the local study population and complied with the Health Insurance Portability and Accountability Act. Written informed consent was obtained from participants age ≥ 18 years or their parents or legal guardians if <18 years.

### 2.2. Dietary Data and Added Sugar Intake

At baseline, 12-, and 60-month follow-up, and outcome study visits, dietary intake was obtained via a food frequency questionnaire (FFQ) for youth aged 10 years and older; the majority of youth completed the FFQ without assistance after receiving staff instruction [13]. The details of the SEARCH FFQ and its validation are described elsewhere [30]. Briefly, the FFQ contained 85 food lines that queried for weekly frequency of consumption, and for average portion size if the food line item was consumed. The nutrient composition of participant diets was derived using the Nutrition Data System for Research (NDSR, Nutrition Coordinating Center, University of Minnesota, Minneapolis MN, Database version 2.6/8A/23), a comprehensive nutrient database of more than 18,000 food products. While the methods used to calculate added sugars in the NDSR database are not published, it is probable that added sugars are derived from the summation of added sugar values from individual ingredients in recipes [24]. For our analysis, added sugars were all carbohydrates (grams per day) from caloric sweeteners that were added during preparation or processing. The caloric sweeteners included: white sugar (sucrose), brown sugar, powdered sugar, honey, molasses, pancake syrup, corn syrups, high-fructose corn syrups, invert sugar, invert syrup, malt extract, malt syrup, fructose, glucose (dextrose), galactose, and lactose. In addition, overall diet quality according to the 2015–2020 Dietary Guidelines for Americans was assessed by a Healthy Eating Index 2015 (HEI) score, which includes 13 components and has a range of values from 0 to 100 [31], with higher scores associated with greater guideline adherence. One of the HEI components is added sugars, which was excluded from these analyses, making the total possible index score 90.

### 2.3. Arterial Stiffness

At the SEARCH Outcome visit conducted between 2011 and 2015, noninvasive measures of arterial stiffness, including pulse wave velocity (PWV) and augmentation index (AIx), were ascertained using a SphygmoCor-Vx device and tonometer. Measurements were obtained in a stable room temperature after 10 min of rest. The utility and details of the arterial stiffness measures in youth including their reproducibility and validity are described elsewhere [32,33,34,35]. Briefly, PWV measurements were obtained at three sites. PWV-trunk measured the pulse transit time from the carotid artery to the femoral artery, is a measure of central arterial stiffness in a large, elastic artery, and it predicts future cardiovascular disease events and mortality [36]. PWV-leg measured the pulse transit time from the femoral artery to the dorsalis pedis artery, which provides a peripheral measure of arterial stiffness in medium-sized, more muscular arteries, and it can be indicative of peripheral vascular disease. Three separate recordings were taken at each site, averaged, and reported in m/s. Higher PWV values indicated increased arterial stiffness. AIx is a measure of wave reflections influenced by central stiffness and also is associated with all-cause mortality in adults [26]. Because AIx is affected by heart rate, values were adjusted to a standard heart rate of 75 beats per minute [33]. Higher AIx values indicated stiffer vessels.

### 2.4. Covariates

At each study visit, questionnaires were used to obtain demographic (birth date, race/ethnicity, highest level of parental education in the household) and diabetes-related information (duration of disease, insulin regimen type, frequency and dosage, and clinical site). Fasting blood samples were obtained under conditions of metabolic stability, defined as no episode of diabetic ketoacidosis during the previous month. Samples were analyzed for glycated hemoglobin (HbA1c), lipid measurements, and glutamic acid decarboxylase-65 (GAD65) and insulinoma-associated-2 (IA-2) diabetes autoantibodies. A Hitachi 917 autoanalyzer (Boehringer Mannheim Diagnostics, Indianapolis, IN, USA) was used for assays of plasma cholesterol, triglyceride, and HDL cholesterol. The Friedewald equation was used to calculate LDL cholesterol if triglyceride concentration was <400 mg/dL (4.52 mM/L) and by the Beta Quantification procedure if triglyceride was ≥400 mg/dL (Hainline). Height, weight, waist circumference, and blood pressure (BP) were measured according to standardized protocol by trained and certified staff. Body mass index (BMI) was calculated as weight (kg)/height squared (m²) and converted to an age and gender-specific BMI z-score [37]. Insulin sensitivity was estimated using an equation validated for youth with diabetes [38], which includes waist circumference, HbA1c, and triglycerides levels. For participants ages ≥10 years, physical activity was assessed using questionnaires from the Youth Risk Behavior Surveillance System (YRBSS) (www.cdc.gov/healthyyouth/data/yrbs/index.htm), and individuals self-reported the number of days per week they participated in physical activity that made them breathe hard or sweat for at least 20 minutes.

### 2.5. Analytic Sample

We included individuals with T1D (combining type 1, type 1a, or type 1b diabetes as assigned by the treating physician), plus a positive diabetes autoantibody test result (GAD65 or IA-2) who had completed the FFQ at the outcome visit (*n* = 1539). Of these eligible individuals, 1517 had PWV trunk data, 1470 had PWV leg data, and 1277 had Aix-75 data, which comprised our sample sizes for cross-sectional analyses.

In addition to the cross-sectional sample at the outcome visit, for some participants (*n* = 553), we had at least two or more measurements of added sugar from the FFQs at the baseline visit and/or follow-up visits. Additional analyses were performed on this subsample to examine the relationship between longer-term estimates of added sugar intake and the outcomes.

### 2.6. Statistical Analysis

Statistical analyses were performed using SAS software (version 9.4; SAS Institute, Cary, NC, USA) [39]. Descriptive analyses were conducted to determine the distribution of demographic measures and arterial stiffness across quartiles of added sugar. Chi-square and analysis of variance (ANOVA) tests were used to compare the quartiles for categorical and continuous data, respectively, with statistical significance established at *P* < 0.05.

Because threshold effects were not evident, separate, multiple linear regression was used to obtain regression (β) coefficients representing the cross-sectional associations between continuous added sugar intake and arterial stiffness. Outcome variables with skewed distributions were log-transformed to improve normality (PWV trunk). For PWV outcomes, Model 1 was adjusted for heart rate, mean arterial pressure, and calories. For Aix-75, Model 1 was adjusted for height, mean arterial pressure, and calories. For all outcomes, Model 2 adjusted for Model 1 covariates, demographic (gender, age, race/ethnicity, and maximum parental education) variables, and diabetes-related variables (diabetes duration and insulin regimen). Model 3 adjusted for Model 2 covariates and overall diet quality using the HEI-2015 excluding the added sugar component. Given evidence [40,41] that the effect of added sugar may be differential according to BMI z-score and physical activity, we examined the potential for effect modification separately using interaction terms and likelihood ratio tests (criterion, *P* < 0.1). For significant interactions, we conducted post hoc tests to examine the effect of added sugars on the outcomes at selected levels of the potential effect modifier. As part of sensitivity analyses, we also examined the interaction of added sugar with HbA1c.

In addition to cross-sectional associations, it was of interest to understand how long-term added sugar intake is associated with arterial stiffness. For the participants with multiple measurements of added sugar, we constructed a longitudinally assessed summary of average sugar intake (i.e., a time-weighted average over all available assessments of added sugar intake). To compute the time-weighted average, we used a stepwise approximation, assuming added sugar intakes were constant for half the time interval before and half the time interval after the visit an intake was reported [42]. We fit similar, separate multiple linear regression models for each outcome using the longitudinally assessed added sugar intake. The same covariate adjustment as for the cross-sectional models was used.

## 3. Results

Our sample consisted of youth with T1D who were predominately non-Hispanic White with a mean age of 18 years (17.7 ± 4.1 years) and had diabetes for 8 years (7.9 ± 1.9 years) at the outcome visit. Mean added sugar intake was 47.0 g/d (interquartile range (IQR) = 26.9–66.4 g/d), and accounted for 12.4% of total calories consumed. Added sugar intake primarily came from sweets and desserts (18.4 ± 16.8 g/d) and sugar-sweetened beverages (17.1 ± 27.4 g/d). Other notable food sources were meal replacement/sports bars (4.1 ± 6.7 g/d), yogurt (3.0 ± 3.9 g/d), and low-fiber grain products (2.2 ± 2.2 g/d).

Demographics and clinical characteristics of the study population at the outcome visit by quartile of added sugar intake are shown in Table 1. Added sugar intake was significantly related to race/ethnicity, maximum parental education, household income, total calorie consumption, and HbA1c. A greater proportion of participants in the highest quartile of added sugar intake were male (compared to females), non-Hispanic Black (compared to non-Hispanic White), and had parental high school education (compared to some college). Participants in the highest quartile also consumed significantly higher calories, had higher HbA1c, and had lower HEI-scores than those in the lowest quartile. No significant associations were observed between added sugar intake and insulin regimen, arterial stiffness measures, or cardiometabolic outcomes.

The multivariate associations between added sugar intake and arterial stiffness measurements after adjustment for covariates are shown in Table 2. In the cross-sectional analysis, added sugar intake was not associated with any measure of arterial stiffness in either the minimally adjusted or fully adjusted models. Although not statistically significant, we converted the β coefficients into more clinically relevant terms by translating grams of added sugar per day into teaspoons of added sugar per day (4 grams per teaspoon). Thus, a β coefficient of 0.0003 for log-transformed PWV-trunk (cross-sectional Model 3) indicates that PWV-trunk increases by 0.1% when added sugar increases by 1 teaspoon per day, and a β coefficient of 0.01 for AIx (cross-sectional Model 2) indicates that AIx increases by 0.05 units when added sugar increases by 1 teaspoon per day.

Given our interest in examining the potentially heterogeneous effects of diet on cardiovascular disease, we examined whether BMI z-score and physical activity modified the relationship between added sugar intake and arterial stiffness measures using our full sample. Multivariate analysis adjusting for potential confounders revealed that the association between added sugar intake and log-PWV trunk was modified by BMI z-score (*P* for interaction = 0.003). Post hoc tests revealed that at higher BMI z-scores (BMI z-score = 2), there was no association between added sugar intake and PWV-trunk (β = −0.0003; *P* = 0.2); however, for lower BMI z-scores, there was a significant, positive association (BMI z-score = 0: β = 0.0005 and *P* = 0.004; BMI z-score = −2: β = 0.001; *P* = 0.0006). To facilitate interpretation on the scale of the original variables, model-predicted log-PWV trunk values were back-transformed at various levels of added sugar intake and displayed by BMI z-score (Figure 1). As shown in the figure, an additional 25 g of added sugar was associated with a 0.16 increase in PWV-trunk when BMI z-score was low (−2) but a nonsignificant increase of 0.05 in PWV-trunk when BMI z-score was high (+2). The 0.16 increase is approximately 13% of a standard deviation in PWV-trunk, which corresponds to a clinically significant mean shift. BMI z-score did not modify the associations between added sugar intake with other measures of arterial stiffness (PWV-arm and AIX-75), and physical activity was not an effect modifier for any measure of arterial stiffness (all *P* for interaction ≥ 0.10). Additional analyses exploring potential effect modifications by glycemic control by including an interaction of HbA1c with added sugar was not significant (*P* for interaction ≥ 0.10).

In the subsample of participants with multiple measurements of added sugar, parallel multivariate analysis revealed similar findings. Long-term added sugar intake was not associated with any measure of arterial stiffness in either the minimally adjusted or fully adjusted models (results not shown). Long-term BMI modified the association between long-term added sugar intake and log-PWV trunk (*P* for interaction = 0.001), but not other measures of arterial stiffness. Long-term physical activity did not modify the association between long-term added sugar intake and arterial stiffness.

## 4. Discussion

In this large cohort of individuals with T1D, added sugar intake accounted for 12.4% of total caloric intake. These findings are fairly consistent, albeit slightly lower, with nationally representative data from the 2005–2010 National Health and Nutrition Examination Surveys (NHANES), which observed that added sugar intake accounted for 16% of total calories in the adolescent (12–19 years) diet [5]. It is possible that youth with diabetes who receive medical nutrition therapy may heed recommendations to monitor carbohydrate intake, although not strictly. The level of consumption among individuals with T1D still exceeds recommendations from the 2015–2020 Dietary Guidelines for Americans [1], the American Heart Association [2], and the American Diabetes Association, which encourages youth with diabetes to minimize consumption of food or beverages that contain added sugar [43]. Our data suggest that the majority of added sugar intake in youth with T1D come from amounts of sugar-sweetened beverages and sweets/desserts. These findings emphasize the importance of identifying barriers to dietary adherence and enablers that help translate nutrition knowledge to behavior, which may be key for improving long-term health in this population.

In our study, the association between added sugar and arterial stiffness was nuanced. There was no association overall between added sugar and arterial stiffness in unadjusted or adjusted models. However, because research suggests that diet may have a heterogeneous effect on CVD risk [44,45,46,47], we a priori hypothesized that the relationship between added sugar and arterial stiffness may vary based on obesity and physical activity. We observed that for individuals with lower BMI z-scores, added sugar intake was positively associated with central arterial stiffness; yet, for those who had higher BMI z-scores, added sugar was not associated with central arterial stiffness in this population of youth with diabetes. The exact mechanism underlying this relationship is not clear, though excess fructose has been shown to increase hepatic de novo lipogenesis and fatty acid synthesis [48]. Previous SNAS results support these findings with higher fructose consumption associated with greater plasma triglyceride levels in youth with T1D [49]. In addition, the effect of sugar on de novo lipogenesis appears to vary by obesity status. Previous studies indicate that in lean individuals, approximately 10% of the fatty acids in very low density lipoprotein (VLDL) cholesterol and triglycerides are attributed to de novo lipogenesis [50]; whereas in obese individuals, this pathway contributes approximately 50% of the fatty acids in VLDL cholesterol [41]. While our data are not consistent with added sugar being deleterious for individuals with higher BMI, we observed that arterial stiffness was significantly higher for those who had higher BMI z-scores than those with lower BMI z-scores regardless of added sugar intake. These findings are consistent with a recent meta-analysis indicating that obese children have increased arterial stiffening, especially in central arteries, in comparison to nonobese children [51]. It is possible for those who are overweight or obese that the excess adiposity supersedes any effect of added sugar on arterial elasticity. Thus, for lean individuals with diabetes, limiting added sugar consumption may be a particularly promising approach for improving long-term cardiovascular health, and for obese individuals with diabetes, weight loss interventions may be more impactful.

Although BMI z-score differentially affected the association between added sugar and PWV trunk, the effect was not observed across all measures of arterial stiffness in our analyses. BMI z-score did not modify the relationship of added sugar with PWV-leg or AIx. The differential associations between added sugar intake and the arterial stiffness measures may be expected given that PWV and AIx are not interchangeable measurements, and obesity may have a greater impact on central arterial stiffness measures than peripheral measures or on Aix [51], which measures arterial stiffness and wave reflection. It is possible that added sugar will not have as much of an effect in more muscular arteries such as the dorsalis pedis artery. Understanding the role of arterial structure on the relationship between added sugar and arterial stiffness is an area of great research potential.

Physical activity is shown to have a cardioprotective effect [52] and is an important factor in maintaining glucose homeostasis [53]. In addition, Bremer et al. observed the interaction of sugar sweetened beverages and physical activity on cardiovascular health in adolescent youth, with low sugar-sweetened beverage intake and high physical activity levels modifying insulin resistance as well as HDL and triglyceride concentrations [40]. Thus, we hypothesized that the relationship between added sugar and arterial stiffness would differ by physical activity status. In our study, we observed that physical activity did not modify the relationship between added sugar and any measure of arterial stiffness in youth with diabetes. It is possible that any modifying effect of physical activity was significant in our sample of youth with diabetes who have higher levels of arterial stiffness [36]. However, Yang et al. found that the relationship between added sugar and cardiovascular disease mortality among U.S. adults 20 years and older was similar between those who had high levels of physical activity and those with low levels of physical activity [11]. Further studies to delineate how physical activity levels interact with dietary intake are warranted.

There are some limitations to our study. Arterial stiffness was only measured at one point in time, precluding a longitudinal analysis. The cross-sectional analysis allows associations to be identified, but not causality or directionality. However, we did examine how long-term added sugar measurements affected arterial stiffness and we observed similar results. SEARCH is an ongoing study, and future analysis will allow a thorough longitudinal examination of the relationship between changes in added sugar intake and changes in arterial stiffness. Other limitations of the study include the use of a food frequency questionnaire, which was validated but relies on retrospective self-recall and is prone to error. In addition, while individuals with diabetes have higher levels of arterial stiffness, it is possible that, given the age of the participants, the impact of diabetes on arterial stiffness was not severe enough to allow us to detect an association; however, we did use continuous measures of both added sugar and arterial stiffness to increase the power to detect an association. We also used BMI z-score as a measure of adiposity in these analyses. While other measures may more accurately capture adiposity, BMI is recommended for large epidemiologic studies and adequately correlates with total body fat [54]. Finally, while we controlled for a number of confounding factors, including overall dietary quality, it is possible that residual confounding affected the results.

Our study also has several important strengths including a large, well-characterized multiethnic sample of youth with T1D who are understudied yet a medically vulnerable population. To our knowledge, this study is the first to assess the relation of added sugar to arterial stiffness in youth with diabetes, and we include multiple measures of arterial stiffness. PWV-trunk measurements predict cardiovascular events above and beyond measurements of traditional cardiovascular risk factors, measure subclinical cardiovascular disease [55], and provide information about cardiovascular risk while the arterial structure is modifiable. In addition, PWV foot measurements provide an understanding of peripheral arterial stiffness in a more muscular artery, which is particularly important, as those with diabetes are at greater risk of peripheral arterial disease [56]. We also capture AIx, which measures both peripheral and central stiffness and captures a part of arterial stiffness not captured by PWV.

In conclusion, our findings indicate that youth with T1D consume similar levels of added sugar to youth in the general U.S. population, which underscores the need to improve dietary adherence in this at-risk population. Further, our study suggests a nuanced relationship between added sugar and CVD risk. Youth with T1D who are lean or less physically active may benefit from a reduction in added sugar consumption. Future studies should explore factors that differentially affect the role of added sugar on CVD and confirm that obesity and physical activity modify this association.

## Figures and Tables

**Figure 1 nutrients-11-01752-f001:**
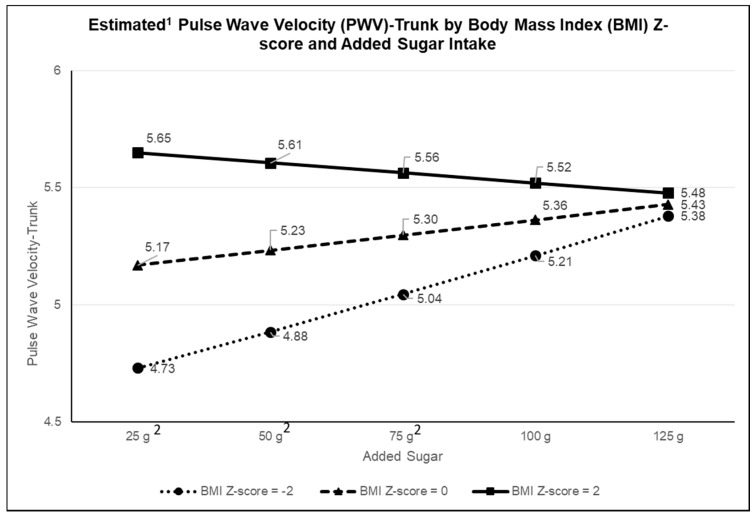
Body mass index modifies the association between cross-sectionally assessed added sugar intake and pulse wave velocity (PWV)-trunk measurements. ^1^ Estimated PWV-Trunk back-transformed from log-PWV trunk, adjusted for heart rate, mean arterial pressure, calories, age at visit, sex, race/ethnicity, parental education, diabetes duration, insulin regimen, and Healthy Eating Index-2015 without added sugar component. ^2^ Pairwise test, *P* value < 0.001.

**Table 1 nutrients-11-01752-t001:** Demographic and clinical characteristics of 1539 participants with type 1 diabetes by quartiles of added sugar: The SEARCH for Diabetes in Youth Study at the Cohort Visit.

Quartiles of Added Sugar Intake					
Variable	Mean (Standard Deviation) or N (%) for All Participants	Quartile 1 (2.2–22.9 g/day)	Quartile 2 (30.0–38.4 g/day)	Quartile 3 (38.5–58.5 g/day)	Quartile 4 (≥58.5 g/day)
n	1539	373	388	386	392
Age at diagnosis	9.8 (3.9; *n* = 1539)	9.8 (3.9)	9.8 (4.2)	9.6 (3.8)	10.1 (3.8)
Age at Outcome Visit	17.7 (4.2; *n* = 1539)	17.7 (4.2)	17.7 (4.4)	17.5 (4.2)	18.1 (3.8)
Diabetes Duration at Outcome Visit	94.6 (22.6; *n* = 1539)	95.1 (22.4)	93.3 (21.9)	94.5 (23.1)	95.6 (23.0)
Race, n (%)^1^					
Non-Hispanic, White	1178	267 (22.7)	294 (25.0)	315 (26.7)	302 (25.6)
Non-Hispanic, Black	145	39 (26.9)	32 (22.1)	25 (17.2)	49 (33.8)
Other	216	67 (31.0)	62 (28.7)	46 (21.3)	41 (19.0)
Gender, n (%)^1^					
Female	795	229 (28.8)	217 (27.3)	186 (23.4)	163 (20.5)
Male	744	144 (19.4)	171 (23.0)	200 (26.9)	229 (30.8)
Highest Parental Education, n (%)^1^					
< High School	60	22 (36.7)	13 (21.7)	11 (18.3)	14 (23.3)
High School Graduate	169	45 (26.3)	43 (25.4)	25 (14.8)	56 (33.1)
Some College Thru Assoc. Degree	484	113 (23.4)	103 (21.3)	145 (30.0)	123 (25.4)
College Degree or More	805	189 (23.5)	222 (27.6)	201 (25.0)	193 (24.0)
Household Income, n (%)^2^					
<$25K	214	52 (24.3)	42 (19.6)	58 (27.1)	62 (29.0)
$25–49K	251	59 (23.5)	56 (22.3)	52 (25.1)	73 (29.1)
$50–74K	234	69 (29.5)	61 (26.1)	49 (20.9)	55 (23.5)
$75K+	578	124 (21.5)	154 (26.6)	164 (28.4)	136 (23.5)
Don’t Know/Refused	254	67 (26.4)	74 (29.1)	49 (19.3)	64 (25.5)
Insulin Regimen, n (%)					
Pump	876	192 (21.9)	22.8 (26.0)	229 (26.1)	227 (25.9)
Long + short/rapid, 3 or more times a day	289	69 (23.9)	74 (25.6)	67 (23.2)	79 (27.3)
Long + any other combo, 2 or more times per day	263	78 (29.7)	68 (25.9)	55 (20.9)	62 (23.6)
Any combo of insulins excluding long, 3 or more times/day	75	25 (33.3)	14 (18.7)	22 (29.3)	14 (18.7)
Any insulin(s) taken 1 time/day, or any insulin combo excluding long 2 times/day	31	8 (25.8)	3 (9.7)	12 (38.7)	8 (25.8)
Total Calories^1^	1681.6 (745.5; *n* = 1539)	1151.2 (377.8)	1476.4 (471.4)	1732.9 (548.0)	2339.1 (887.9)
Pulse Wave Velocity (PWV) Trunk	5.5 (1.2; *n* = 1517)	5.5 (1.2)	5.4 (1.1)	5.4 (1.3)	5.5 (1.0)
Pulse Wave Velocity Leg	8.1 (1.6; *n* = 1470)	8.0 (1.3)	8.0 (1.4)	8.1 (1.4)	8.1 (1.3)
Aix-75	−2.7 (10.5; *n* = 1331)	−1.4 (9.8)	−3.1(10.8)	−2.8 (10.2)	−3.4 (11.1)
Mean Arterial Pressure	81.1 (9.1; *n* = 1424)	81.2 (8.4)	80.2 (8.8)	81.0 (8.7)	81.9 (10.1)
Heart Rate, mean % (SD)	68.6 (11.39; *n* = 1316)	69.6 (11.4)	69.4 (12.0)	68.6 (10.8)	67.6 (11.4)
BMI z-score	0.62 (0.93; *n* = 1316)	0.7 (1.0)	0.6 (0.9)	0.6 (0.9)	0.6 (0.9)
Weight Status, n (% overweight/obese)	539	147 (27.3)	125 (23.2)	136 (25.2)	131 (24.3)
Weight Status, n (%obese)	202	58 (28.7)	42 (20.8)	58 (28.7)	44 (27.8)
Physical Activity, n (% active 3–7 days/week)	888	203 (22.9)	224 (25.2)	233 (26.2)	228 (25.7)
HbA1c, mean % (SD)^1^	9.12 (1.82; *n* = 1305)	9.0 (1.7)	9.0 (1.8)	9.1 (1.7)	9.5 (2.0)
Waist Circumference	84.5 (13.5; *n* = 1314)	84.9 (14.3)	83.2 (13.5)	83.6 (13.1)	84.0 (12.6)
Triglycerides, mean mg/dL (SD)	92.6 (71.7; *n* = 1465)	92.0 (59.2)	92.0 (72.1)	94.5 (87.7)	91.9 (64.3)
Log-Insulin Sensitivity score	1.83 (0.4; *n* = 1486)	1.8 (0.4)	1.9 (0.4)	1.8 (0.4)	1.8 (0.4)
Systolic blood pressure, mean mmHg (SD)	106.3 (10.9; *n* = 1538)	106.5 (11.0)	105.4 (10.5)	106.5 (11.1)	106.8 (10.8)
Diastolic blood pressure, mean mmHg (SD)	68.6 (8.8; *n* = 1538)	68.9 (8.8)	68.0 (9.3)	68.5 (8.7)	69.0 (8.7)
Healthy Eating Index (HEI)-2015 Total Score (without added sugar component)^2^	47.7 (10.7; *n* = 1539)	48.2 (11.8)	48.8 (10.6)	47.1 (10.4)	46.9 (10.0)

^1^*p* < 0.01, ^2^
*p* < 0.05.

**Table 2 nutrients-11-01752-t002:** Multivariate regression models between cross-sectionally and longitudinally assessed added sugar intake (area under the curve estimate of average added sugar intake over the study duration, 2–12 years) with arterial stiffness measures in youth with type 1 diabetes (T1D).

**Cross Sectional Assessed Added Sugar Intake (Regression [β] Coefficient ± Standard Error)**			
	Log PWV^1^-Trunk	PWV^1^-Leg	AIx^2^
Model 1^3^	0.0002 ± 0.0002	−0.0001 ± 0.001	0.02 ± 0.01
Model 2^4^	0.0003 ± 0.0002	−0.000005 ± 0.001	0.01 ± 0.009
Model 3^5^	0.0003 ± 0.0002	−0.0001 ± 0.001	0.01 ± 0.01
**Longitudinally Assessed Added Sugar Intake (Regression [β] Coefficient ± Standard Error)**			
	Log PWV^1^-Trunk	PWV^1^-Leg	AIx^2^
Model 1^3^	0.00002 ± 0.002	−0.0006 ± 0.001	0.02 ± 0.01
Model 2^4^	0.00008 ± 0.002	−0.0007 ± 0.001	0.02 ± 0.01
Model 3^5^	0.0001 ± 0.0002	−0.0009 ± 0.001	0.02 ± 0.01

^1^ PWV – Pulse Wave Velocity. ^2^ AIx – Augmentation Index. ^3^ For pulse wave velocity outcomes, Model 1 was adjusted for heart rate, mean arterial pressure, and calories. For augmentation index, Model 1 was adjusted for height, mean arterial pressure, and calories. ^4^ Adjusted for Model 1 covariates, age at visit, sex, race/ethnicity, parental education, diabetes duration, and insulin regimen. ^5^ Model 2 covariates, plus HEI-2015 score without added sugar component.

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
