# Peer review of "Body Mass Index Z-Score Modifies the Association between Added Sugar Intake and Arterial Stiffness in Youth with Type 1 Diabetes: The Search Nutrition Ancillary Study"

_nutrients, 2019, doi:10.3390/nu11081752_

Reviewer 1 Report

The manuscript entitled “The Association between added sugar intake and arterial stiffness in youth with Type 1 Diabetes: the SEARCH Nutrition Ancillary Study” This is an interesting observational study on the authors have been analyzed the relationship between sugar intake and arterial stiffness in young individuals with T1D. The authors have also evaluated the implication of other variables such as the BMI value and physical activity and after a rigorous statistical analysis the conclusions obtained determined that only in those patients with lower BMI z-score, sugar intake was positively associated with a higher arterial stiffness determined by the PWV value.

Please find below minor comments associated with this manuscript.

1.      The authors should include a reference of SAS software.

2.      For the analysis of sugar intake, has separated the analysis between the intake of simple sugars or easily metabolizable and complex sugars?

3.      The authors can describe how they get and what is the b-coefficient

Reviewer 2 Report

This manuscript described the effect of added sugar intake among youth with type 1 diabetes on arterial stiffness measures. More specifically, investigating BMI z-sore and added sugar on pulse wave velocity trunk measurements.

Comments:

Abstract

·         The abstract does not include the overall association between added sugar intake and arterial stiffness measures. Although not significant, I believe it should be reported in the abstract before discussing the BMI z-score results.

Introduction

·         Consumption of sugar sweetened beverages and CVD risk factors was introduced. However, the only statement about arterial stiffness consisted of “no effect on arterial stiffness”. It would be important to include any papers discussing stiffness in adults/youth with added sugar. Otherwise, it seems out of place to propose that you will see any effect if there is no data to support the hypothesis. The introduction focused on blood lipid measures and sugar intake and nothing linking these to arterial stiffness.

Methods

·         Galactose and lactose are listed as caloric sweeteners. Do dairy products that naturally contain these sugars account for any of the added sugars?

·         It would be nice to see details about the added sugar intake in a table. Only the mean intake with the IQR was given. What were the top foods that made up the added sugar intake? Was this sweets, sweetened dairy products, SSB, pastries etc., and/or was it a wide variety across individuals?

·         What time of year was each FFQ taken? Also, was the questionnaire completed by the children or caregiver?

·         Is there any reference for validation of the arterial stiffness measures in children?

·         Why does the amount of participants decline across the stiffness measure indices?

·         How was physical activity assessed?

Results

·         How many more FFQ assessments were done in the subset analysis group?

Discussion

·         It is described that the findings of added sugar intake that exceeds the dietary guidelines “emphasize the importance of identifying barriers to dietary adherence and enablers”. I feel that it would put the exceedingly high intake into perspective if actual foods that contain added sugar of the cohort you studied were described. Even small overindulgences for a child would account for 25g of sugar (e.g. 1.5 chocolate chip cookies, 1 candy bar, 1 flavored yogurt, 1.5 scoops of ice cream, 1 small sports drink, 1 sweetened milk drink etc.). Children with type 1 diabetes have a life-long commitment to monitor their blood sugar and administer insulin and having to scrupulously restrict added sugars must be extremely difficult. This could possibly be addressed in the discussion.

·         Measures of blood lipids among this SEARCH trial was not referenced in the discussion with such an emphasis on blood lipids and CVD risk in the introduction some of this information may be valuable for the discussion. (Diabetes Care 29:1891-1896, 2006)

·         The conclusion that this study “suggests a nuanced but deleterious relationship between added sugar and CVD risk” is too strong of a statement given the nature of this study design and limitations.  

Overall, this manuscript provided insight, in that regardless of BMI status, children with type 1 diabetes do not show and association between added sugar intake and arterial stiffness. This hold true given that the group studied consumed 12.4% of their total calories from added sugars. Moreover, the participants with a lower BMI z-score had a worsening of PWV trunk measures with higher intake of added sugar. The major limitations of this paper include the assessment of dietary intake that was limited to 3 FFQ and the statistical analysis assumes added sugar intakes were constant based on those 3 assessments over a 5 year period. Additionally, adiposity was crudely measured by BMI and not actual measurements of body fat. Waist circumference measures would provide more insight into adiposity than BMI. Finally, physical activity was assessed to see if it modified the effect and the methods for physical activity assessment were not detailed.

Reviewer 3 Report

The title of the paper did not directly seem to reflect the claim and conclusion of the paper. All results did not clearly produce a scientifically interesting message. It might be related to the one-point measurement of the outcomes. This did not make a right way to a define conclusion. Or the changes of outcomes were not still seen in the youth generation even in diabetes.

Author Response

Reviewer 3

The title of the paper did not directly seem to reflect the claim and conclusion of the paper. All results did not clearly produce a scientifically interesting message. It might be related to the one-point measurement of the outcomes. This did not make a right way to a define conclusion. Or the changes of outcomes were not still seen in the youth generation even in diabetes.

We have updated the title of the manuscript to reflect that BMI z-score modifies the relationship between added sugar and arterial stiffness in youth with type 1 diabetes.

We appreciate the reviewer’s concern. We agree that it is challenging to interpret results that show a non-significant finding overall, but that BMI-z score modified these findings. However, we have chosen to report the results of our a priori analysis based on our a priori hypothesis. We feel that to maintain scientific integrity, it is essential to present all results even with nuanced findings. There is an increasing number of research studies that show conflicting nutrient-health relationships which may be attributed to effect modification of various factors including adiposity, physical activity, and other epigenetic/microbiome effects, which supports the value of us examining effect modification. 

We agree with the reviewer’s concern that a limitation of the paper is that the outcome has only been measurement at one point in time. This cross-sectional analysis prohibits understanding whether the relationship is causal; however, we do not have reason to suppose that arterial stiffness would affect added sugar intake. Further, we address limitations of cross-sectional analysis by creating a historical measurement of added sugar by creating a time-weighted average. Limitations of these analyses are addressed in the discussion section (page 9).

We appreciate the reviewers concern that we have a younger population. Although our participants have diabetes and have higher levels of arterial stiffness, it is possible that the arterial stiffness is not severe enough (or consistent with levels observed in later adulthood) to detect an association between added sugar and arterial stiffness. We have added this to the limitation section of our discussion section in the revised manuscript.

Round  2

Reviewer 3 Report

Describe the ‘standard’ beta-coefficient in Table or within Text. How did the BMI-Z modify the beta level? Based on the results, the authors discuss the results. P for interaction could easily show a slight significance when analyzing it with the large sample size. This is significant but not relevant. The discussion may be more modest.
